# Study on the Classification Performance of Underwater Sonar Image Classification Based on Convolutional Neural Networks for Detecting a Submerged Human Body [note 1]

**DOI:** 10.3390/s20010094

**Published:** 2019-12-23

**Authors:** Huu-Thu Nguyen, Eon-Ho Lee, Sejin Lee

**Affiliations:** 1Department of Mechanical Engineering, Kongju National University, Cheonan 31080, Korea; nguyenhuuthu.vmu@gmail.com (H.-T.N.); Eonho@smail.kongju.ac.kr (E.-H.L.); 2Department of Mechanical and Automotive Engineering, Kongju National University, Cheonan 31080, Korea

**Keywords:** image classification, underwater multi-beam sonar, sonar noise, convolutional neural network, human body detection

## Abstract

Auto-detecting a submerged human body underwater is very challenging with the absolute necessity to a diver or a submersible. For the vision sensor, the water turbidity and limited light condition make it difficult to take clear images. For this reason, sonar sensors are mainly utilized in water. However, even though a sonar sensor can give a plausible underwater image within this limitation, the sonar image’s quality varies greatly depending on the background of the target. The readability of the sonar image is very different according to the target distance from the underwater floor or the incidence angle of the sonar sensor to the floor. The target background must be very considerable because it causes scattered and polarization noise in the sonar image. To successfully classify the sonar image with these noises, we adopted a Convolutional Neural Network (CNN) such as AlexNet and GoogleNet. In preparing the training data for this model, the data augmentation on scattering and polarization were implemented to improve the classification accuracy from the original sonar image. It could be practical to classify sonar images undersea even by training sonar images only from the simple testbed experiments. Experimental validation was performed using three different datasets of underwater sonar images from a submerged body of a dummy, resulting in a final average classification accuracy of 91.6% using GoogleNet.

## 1. Introduction

### 1.1. The Necessity of Submerged Body Detection

The techniques of recognizing an object in turbid water and making a decision on the type of the object are essential factors for a diver or a submersible to support marine tasks [1,2].

For that, a sonar image and image processing techniques have been used to detect and identify underwater objects at the seafloor level [3]. Cho et al. proposed a method to provide an automatic detection alarm indicating the presence of suspected underwater objects using a high-speed imaging sonar. This alarm signal alerts human operators or automated underwater vehicles to suspected objects, which may be a part of or all of the target object [4]. REMUS 6000 autonomous underwater vehicles (AUVs) had successfully searched for Air France Flight 447 which crashed in 2009 [5]. Reed et al. introduced the method of mine detection and classification using high-resolution side-scan sonar for mine counter measures (MCM) [6].

Especially when it comes to finding somebody in the water, there are various requirements. The first is when a person is missing in the water. To find bodies in the water, it is important to search soon after the accident occurs. Also, it is crucial to ensure the safety of the rescuers. However, harsh on-site conditions, such as high flow rates, poor water visibility, or the presence of dangerous obstacles in the water, do not guarantee the safety of rescuers. The second is the case of the defense dimension. If the enemy penetrator tries to infiltrate underwater, it should be able to monitor and detect it at the defense level effectively. The efficiency of the operation, such as discrimination and judgment, is low due to the extreme fatigue because the operator has to be continuously observing the optical or sonar image. There are few such studies of how to automatically detect a submerged body using an underwater sonar image captured in poor water conditions [7,8].

For the vision sensor, the water turbidity and limited light condition make it difficult to take high quality image as shown in Figure 1. For this reason, sonar sensors are mainly utilized in water [9,10]. However, even though a sonar sensor can give a plausible underwater image in the limitation, the quality of the sonar image varies greatly depending on the background of the target [1]. The readability of the sonar image is very different according to the target distance from the underwater floor or the incidence angle of the sonar sensor. The target background effect should be considered to overcome because it causes scattered noise or polarization noise in the sonar image.

### 1.2. Deep Learning

Machine learning techniques are used to identify objects in an image, copy voice to text, match new items, posts, or products to the user’s interests, and select relevant results of the search. This technology has recently evolved into a technique called deep learning of more complex structures [12]. Existing machine learning techniques have limited ability to process critical information in raw data. To build a pattern recognition or machine learning system, considerable expertise is needed to design a feature extractor that converts raw data to the appropriate representation or feature vector. In contrast, recent deep learning is a method that automatically detects the expressions needed for detection or classification using the provided raw data. The deep learning method is a simple but nonlinear module to transform the expression at the natural input level to a higher and more abstract representation.

For this reason, in this study, we tried to find the shape of the submerged body in the sonar image by using the deep learning method with a considerable level of noise. The CNN-based models such as AlexNet [13] or GoogLeNet [14] are two types of artificial neural network that has been studied extensively in various image processing and computer vision fields, such as understanding images and extracting high-level abstract information from them or drawing pictures with new textures. The fundamental CNN has four key ideas that demonstrate the benefits of natural signal characteristics, including local connectivity, shared weights, pooling, and the use of many layers. The general architecture of a CNN consists of a series of stages of the convolution layer and the pooling layer. These CNNs learn to extract features from their images in their own rules by repeating learning and then generate unique maps, and finally, connect layers similar to existing hierarchical neural networks to produce desired results.

### 1.3. Paper Contents

In a previous study, we proposed the method to automatically classify images containing submerged bodies by using underwater multi-beam sonar images without a noise effect [15]. In this study, we propose a method to automatically classify images with noise effects such as background and polarizing noise. Notably, to automatically classify the images including a submerged body, these sonar images are learned using CNN-based models such as AlexNet and GoogLeNet among deep learning methods. To successfully classify the sonar image in the real sea area after training the sonar image from the testbed, we prepared the training data and the test data by using the adequate image processing techniques. There are three different datasets of underwater sonar images which contain a submerged dummy. One dataset was captured in the very clean water testbed of Korea Institute of Robot and Convergence(KIRO), two other datasets were taken under the very turbid water condition on Daecheon beach in Korea in 2017 and 2018, we named them CKI, TDI-2017 and TDI-2018 sets prospectively. CKI dataset was utilized as the training data, and TDI-2017 and TDI-2018 datasets were applied for the test data of deep learning. Two labels for classification were set up such as body and background. Then, the training data was learned by AlexNet or GoogLeNet, and its classification performance is confirmed from two labeled test sets depending on whether or not a submerged body exists in the image. In Figure 2, we present an image-processed training & test data specifically for underwater sonar images. This study focused on as follows:Experiments were carried out to acquire underwater sonar images for the study of the submerged body detection method. Mainly, we obtained the experimental data in both clean and turbid water environments.Through the case study to validate the robust submerged body detection, the reasonable classification performance of sonar images including a submerged body was obtained using a CNN-based deep learning model.The most important thing in this study is to confirm the feasibility of applying a deep learning model only with CKI to a field image of the seaside. For that, we prepared the training data by using image processing techniques that can realize general and polarized noise by background. With this possibility, indoor testbed data alone will provide a starting point for considering various field robotic applications.

The remainder of this paper is organized as followings. Section 2 gives a brief introduction of CNN-based models. Section 3 explains the utilization of image processing techniques to successfully classify the sonar image in the real sea area after learning the sonar image from the testbed. Experimental results are given in Section 4; finally, discussion and conclusions are drawn in Section 5.

## 2. Convolutional Neural Network

CNN is a feed-forward neural network that is applied to image recognition. In the former feed-forward neural network, all the units in adjacent layers were fully connected to each other. On the contrary, CNN has a special layer in which only a specific unit between adjacent layers has a bond. At this layer, the basic operations of image processing such as convolution and pooling occur. It was inspired by neuroscientific knowledge of the visual cortex in the brain. Currently, multiple layers of CNN are becoming the most important technology in image recognition problems in general. Object category recognition, which identifies the type of objects contained within a single image, was easy for people but difficult for computers. CNN has solved these long standing problems. In the field of neuroscience, electrophysiological experiments have revealed that the CNN behaves like a high dimensional visual cortex existing in the brain of a primate [16]. Among many Computer Vision’s (CV) technologies, CNN-based technologies have been used by many people for it’s better performance to compare with the existing CV technologies. That performance resulted in Imagenet competition, the most famous and important competition which is held annually since 2010 to evaluate algorithms for object detection and image classification. Since the results of imagenet 2012, CNN technology is used a lot in image classification, other CNN-based architectures have been also continuously developing and achieving excellent results in the ImagetNet competition like GoogLeNet (2014), ResNet [17] (2015). When CNNs had not been used often for image classification, people used to create features from images and then feed those features into other classification algorithms. While CNNs can automatically extract the image’s features. Achieve high accuracy on difficult classification tasks, flexible abilities and perform well on images, we chose CNN-based methods in our model for the purpose of classification.

### 2.1. CNN Architecture

The CNN structure is often used in image recognition, such as object category recognition. From the input side to the output side, the convolution layer and the pooling layer are paired in a form in which the convolution layer comes first and then the pooling layer comes out, and this pair is repeated many times again. However, there are cases in which convolution layers are repeated before followed by a pooling layer. After the convolution layer and the pooling layer are repeated, a layer in which the unit between adjacent layers is fully connected is disposed of. This is called the fully connected layer and distinguishes it from the convolution layer. The fully connected layer is generally arranged by connecting several layers. The final output layer uses a softmax layer for classification purposes.

### 2.2. Definition of Convolution Layer

Considering a gray-scale image storing intensity values in each pixel, the image size is assumed to be W×W and the pixel index is represented by (i,j). The pixel value is $x_{ij}$. Assume a small image called a filter, and let it be H×H in size. The pixel of the filter is represented by index (p,q), and the pixel value is written by $h_{pq}$. The convolution of an image is the following operation defined between an image and a filter.
(1)$$u_{ij}=\sum_{p=0}^{H-1}\sum_{q=0}^{H-1}x_{i+p,j+q}h_{pq}$$

### 2.3. AlexNet

The ImageNet Large Scale Visual Recognition Competition(ILSVRC)-2012 winner based on the ImageNet database was won by Alex Khrizevsky of the University of Toronto, Canada. From his name, the CNN structure they developed is called AlexNet. AlexNet is not very different from LeCun’s LeNet5 when viewed from a structural point of view, but it has potential application in a lot of places to get high performance. Especially since GPU has very meaningful results, the use of GPUs has become a popular trend when designing CNN structures.

AlexNet consists of five convolution layers and three fully-connected layers, and the last fully-connected layer uses the softmax function as an activation function for classification into 1000 categories. AlexNet has a vast CNN structure with about 650,000 neurons, 60 million free parameters, and 630 million connections, and uses two GPUs for learning about this vast network.

### 2.4. GoogLeNet

In 2014, ILSVRC was occupied by Google’s GoogLeNet, and VGGNet from Oxford University was ranked second only to a very small extent. AlexNet, ZFNet and the original LeNet5 are quite simple compared to the structure in 2014 and their network depth is less than ten layers. However, CNN’s structure has begun to change dramatically since 2014. The most direct way to improve CNN’s performance is to increase the size of the network. Increasing the size of a network simply means increasing the number of units in each layer as well as increasing the number of layers in the network. Especially, it is almost necessary to learn using a large amount of data like ImageNet data. GoogLeNet and VGGNet became deeper with 22 layers and 19 layers, respectively. Of course, the top-5 error rates also fell to 6.7 percent and 7.3 percent, respectively.

Increasing the size of the network can improve performance, but if not, you may encounter two serious problems. First, as the network grows larger, the number of free parameters increases and the network becomes more likely to fall into over-fitting. Another problem is that as the network size increases, the number of computations increases. Therefore, it is not just making the network deeper, but it requires some structural consideration. Accordingly, Google developed the basic structure of Inception. This is achieved by applying convolution filters of different sizes to the same layer to obtain different scale features. The 11 convolution filter is appropriately used to reduce the dimension and solve the problem of increasing the amount of computation when the network is deepened.

The most important part of GoogLeNet is that the depth of the network is much deeper compared to AlexNet, the number of free parameters is about 1/12 and the total number of computations is smaller. GoogLeNet has 9 Inception modules.

## 3. Image Pre-Processing for Deep Learning

### 3.1. Uncommon Sensor Images

Generally, photographic images are used to perform image classification or object detection using deep learning. As a representative example of this, there is ImageNet competition. In 2009, Fei-Fei Li unveiled 15 million image databases that categorize objects and objects into 22,000 categories. For example, in the case of cats, the image net database included 62,000 images of all shapes, attitudes, and everything from house cats to hand-picked. And this massive image database was made available for free to everyone. The primary purpose of training the deep learning model using photographic images is to classify objects and recognize objects or environments in most situations. It can lead to various universal applications such as autonomous vehicles. In this case, generalization tasks such as regularization, weight attenuation, and dropout are essential to prevent the deep learning model from being trapped in the local minimum situation. And, as in the cat example mentioned above, the use of images in various situations for the same category is significant. The reason for this is to ensure that the cat’s image applied at the test is robust enough to answer the cat, no matter what kind, shape or color of cat it is.

However, as in this study, images of specific environments or situations need to be different in terms of the research approach. We tried to solve the classification problem by using a multi-beam sonar image to detect the human body underwater. Although there are no sensors that can provide as accurate information as sonar images in turbid water, multi-beam sonar images are basically inferior in resolution, and the ability to express the shape of an object is feeble due to the nature of its sensing mechanism. And, unlike the case of a cat, there are no various environments or situations for a particular object underwater. Therefore, rather than focusing on the diversity of the training image, it is desirable to improve the classification accuracy through some important transformations of the training image.

### 3.2. Noise Generation

In this study, background and polarizing noise are added to the original CKI, the intensity is controlled, and the inversion effect is applied to augment the training image. First, the reason for adding noise is that the object can be placed on the underwater floor. In CKI, there is no noise around the object because the dummy suspends in the water. On the other hand, in the TDI, because the dummy is located on the seabed, there is a noise phenomenon due to ultrasonic reflection from the floor as well as the dummy. The noise phenomenon on the ground varies depending on the heading angle of the sonar sensor with respect to the underwater surface. As the heading direction of the sonar sensor faces the normal vector of the ground, the background noise becomes more polarized. On the other hand, as the heading direction of the sonar sensor deviates from the normal vector of the ground, the phenomenon that the background noise spreads widely becomes noticeable. We implemented this noise phenomenon not only by spreading the noise extensively on the original CKI but also by applying the polarizing effect as shown in Figure 3 and Figure 4.

Next, the pixel intensity was adjusted also as shown in Figure 3. The object placed on the underwater floor differs in sharpness according to the inclined angle of the sonar sensor. Therefore, we tried to realize the difference of sharpness that can be seen by adjusting the pixel intensity. Finally, the black and white inversion effect was applied as shown in Figure 5. Since the distance between the sonar sensor and the dummy was somewhat close to that of the actual TDI-2017 acquisition, the pixel intensity around the dummy was strong, and the pixel intensity concerning the bottom was relatively low. However, the distance between the sonar sensor and the dummy was relatively longer in the TDI-2018 acquisition experiment than in the 2017 experiment. As a result, the ultrasonic waves were absorbed from the dummy surface, and the bottom surface was bright in the image, but only the portion where the dummy was present looks empty. Therefore, we wanted to implement both styles through image inversion.

### 3.3. Image Preparation for Training

Figure 3 shows how the original image changes with the addition of noise. Figure 3a shows the result of applying a noise density of 0.3–0.6 with ‘salt & pepper’ effect to a sample of the original image of CKI. Figure 3b shows the result of blurring the image of Figure 3a with a Gaussian filter with a standard deviation of 2. Figure 3b also shows the image when the pixel intensity changes from 1.6 to 2.0, and Figure 5 shows the resulting image when the image of Figure 3b is inverted.

As a result, one source image finally generates 60 images through five cases in noise addition, three cases in pixel intensity variation, two cases in reversal effect, and two cases of polarizing application. All this image processing was done using Matlab. The used functions were ‘imnoise (’salt & pepper’)’ for adding noise and ‘imgaussfilt()’ for Gaussian blurring. The pixel intensity was obtained by multiplying the original image pixel by 1.6 and 2.0, and the inverse effect was implemented by subtracting the image pixel value from the maximum value of 255 for each pixel.

## 4. Experimental Results

### 4.1. Image DataSets

There are three different datasets of underwater sonar images including submerged body shapes caused by a dummy were used. One dataset named CKI what stands for clean KIRO images, was captured in the very clean water testbed of Korea Institute of Robot and Convergence (KIRO). The maximum depth of this water testbed is as much as about 10 m, a dummy positioned in a water depth of about 4 m was used to effect the submerged body as shown in Figure 6. TELEDYNE BlueView M900-90 mounted on a gearbox capable of rotating the sonar sensor at the angular interval of 5 degrees was used to obtain efficient underwater sonar images from various angles. BlueView M900-90 is a multi-beam imaging sonar in which the field of view is 90 degrees, the beam width is 20 degrees, the sonar frequency is 900 kHz, and the maximum range is 100 m. For artificial neural network learning, we cropped 31 images containing the submerged body and empty area, respectively. These images were used as original images to make augmentations with noise for the training of deep learning. Figure 7 shows some CKI examples. The training image set was classified into three types according to the noise addition condition such as ‘Original-CKI’, ‘Background noised-CKI’, ‘Background & Polarizing noised-CKI’. Original-CKI set includes only images augmented according to the level of intensity and the inversion. Each label has 186 images. Background noised-CKI set includes images augmented according to the level of intensity and ‘salt & pepper’ effect and the inversion. Each label has 930 images. Background & Polarizing noised-CKI set includes images augmented according to the level of intensity and ‘salt & pepper’ effect, the inversion, and the polarizing effect. Each label has 1860 images.

Two other datasets for the classification test, named TDI-2017 and TDI-2018 what stands for turbid Daecheon beach images, were collected on Daecheon beach with the very turbid water in west Korea in 2017 and 2018 prospectively. On the west coast of Korea, seawater is very cloudy and is also called the Yellow Sea. The BlueView M900-90 was fixed to the lower part of the kayak as shown in Figure 8, and the heading direction was mounted at about 30 degrees downward from the water surface. The biggest difference between the experiments to obtain TDI and CKI datasets is about the position of the mannequin in the water. The mannequin was suspended in the water in CKI experiment while it was located at the seabed when collecting TDI datasets. Another thing is that the mannequins were dressed. In the experiments, the distances between the sensor and the dummy were about 2–4 m for the TDI-2017 and 4 to 10 m for the TDI-2018. For artificial neural network testing, we cropped every 30 images containing the submerged body and background area from TDI-2017 and TDI-2018 respectively.

### 4.2. Model Setup and Learning

The computer system’s operating system for learning CNN-based AlexNet and GoogLeNet is Ubuntu 14.04 LTS, Memory 64 GB, Processor Intel Core i7-5930K and 4 GPUs of GeForce GTX 1080. We used Nvidia’s DIGIT 6.1 with Torch framework as the training system. The learning models consisted of six classes, Original-CKI trained AlexNet and GoogleNet, Background noised-CKI trained AlexNet and GoogleNet, and Background & Polarizing noised-CKI trained AlexNet and GoogleNet. A total of 15 epochs were performed. The learning rate was set to 5e${}^{-6}$. The input image size is converted to 256×256 pixels and the image type is set to Gray-scale. Adaptive Moment Estimation(Adam) was used as a solver for back-propagation update. The options of Flipping, Quadrilateral Rotation, Rotation(+− deg), Rescale(stddev), and Noise(stddev) were all activated.

### 4.3. Classification Results

In this study, two kinds of labels as body and background were set for sonar image classification. The training datasets were prepared in three types: original CKI, background noised CKI, background & polarizing noised CKI. Figure 9 and Table 1 show the classification results for six cases. All cases were expressed as the product of the classification accuracy for every 30 images of the body and background labels. Both AlexNet and GoogleNet models that trained with original CKI can see that there was no classification performance no matter what the model is because the average classification accuracy was lower than 50% ( 43.3% and 31.6% prospectively). In the model trained with background noised CKI, GoogleNet’s result with respect to TDI-2018 showed classification accuracy of 80%, but AlexNet did not work at all (9.6% ). On the other hand, with background and polarizing noised CKI, the classification performance of GoogleNet model was steady at 63.8% with both data in 2017 and 2018. In other cases, the classification accuracy varied slightly depending on the type of label or dataset with medium classification accuracy.

The image classification results were labeled with a probability from 0% to 100%. So after testing, we could compare the classification results from CNN with the hand-made ground truth to check whether it succeeded or failed. Figure 10 shows the sample images of (a) TDI-2017 and (b) TDI-2018 that failed to be classified into the label of background. The common feature of Figure 10a is that the background pattern of the coast is relatively enlarged because it crops out a rather small area while Figure 10b are images of background with a shadow of uneven seabed that caused the failure of classification into the label of background. Figure 11 shows the sample images of (a) TDI-2017 and (b) TDI-2018 that failed to be classified into the label of body. Figure 11a are images of situations where Polarizing phenomena are very strong. In the images of Figure 11b, the overall size of the body is rather small than the whole image size and the body area is a bit hazy.

### 4.4. Re-Training

To receive better results, the model needed to be re-trained. The polarizing and intensity were tested at three levels from 1 to 3 with GoogleNet. Figure 12 shows sample images according to polarizing and intensities. After retraining, the results were so much better. Both TDI datasets in 2017 and 2018 received better and more stable classification accuracy with all three levels as shown in Figure 13 and Table 2. At level 3 of polarizing and intensity, the classification accuracy was highest which is 100% with TDI-2018 dataset and 91.6% on average.

## 5. Discussion and Conclusions

In this study, we confirmed that we can classify images captured from the ocean using only sonar images obtained from the testbed in training the deep learning model. However, the sonar images obtained from the testbed and the seaside are very different from each other, so that if the images obtained from the testbed were trained as they are, the ocean images were not classified at all. We confirmed that seaside images can be classified well by adding background, intensities, blurring, inversion, and polarizing noise that can be observed in marine sonar images to the original image of the testbed. From this fact, we can find out what we can put in the field immediately after training in the testbed when we want to detect a specific object in the ocean in the future. To confirm this possibility, we experimented with actual ocean sonar images through two experiments. And we applied the CNN-based deep learning model called GoogleNet. The article result has a huge range of applications for not just finding a submerged human body for safety reason, applying in an emergency rescue or military service especially in the underwater surveillance system that could detect divers or other underwater intrusions, like underwater vehicles but also be appropriate to apply in other fields such as underwater archeology or fishing industry for detecting sea creatures. As future research, we will carry out a study on object detection using underwater sonar images. 

## Figures and Tables

**Figure 1 sensors-20-00094-f001:**
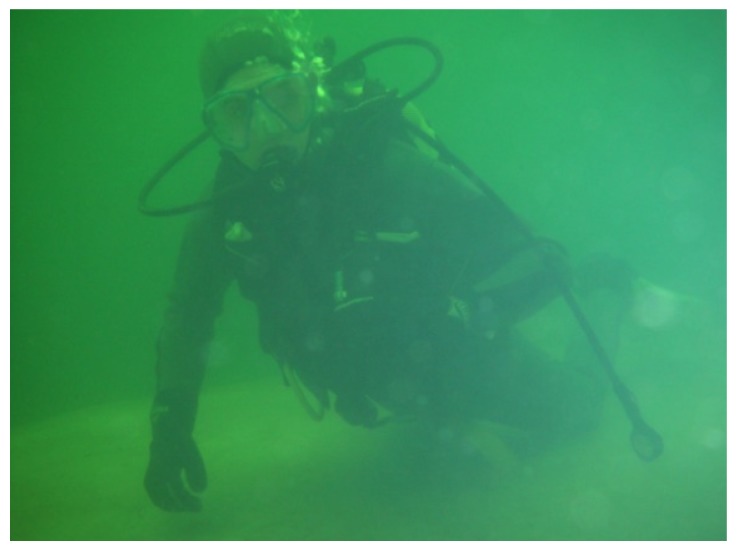
An example picture of poor visibility in turbid water [11].

**Figure 2 sensors-20-00094-f002:**
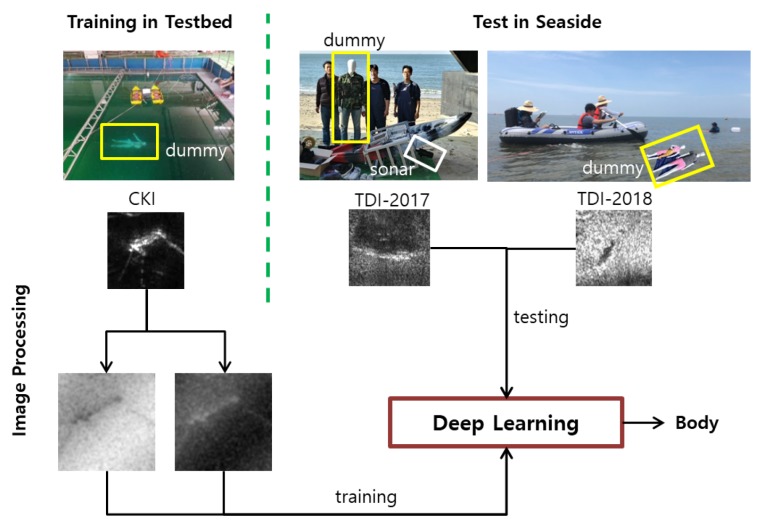
Overview of the proposed approach. We trained the model using CKI with image processing techniques and tested real seaside images of TDI-2017 and TDI-2018.

**Figure 3 sensors-20-00094-f003:**
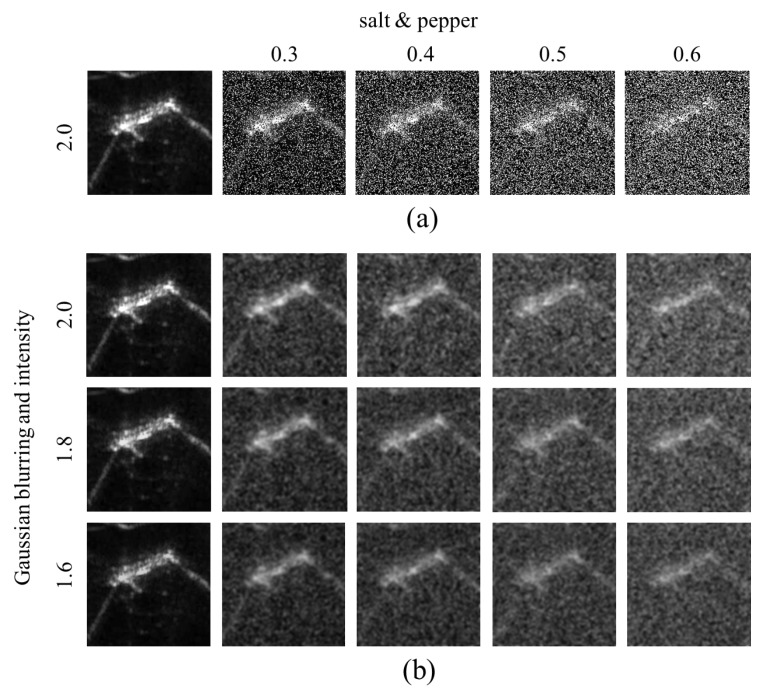
Images according to the noise density of ‘salt & pepper’ effect, Gaussian blurring, and pixel intensity for the training dataset. The background of the target caused noise to the sonar image, so ‘salt & pepper’ noise to the CKI was applied in (**a**) and then blurred in (**b**). The image sharpness of (**b**) was varied by adjusting the pixel intensity because the sharpness can be differed according to the inclined angle of the sonar sensor.

**Figure 4 sensors-20-00094-f004:**
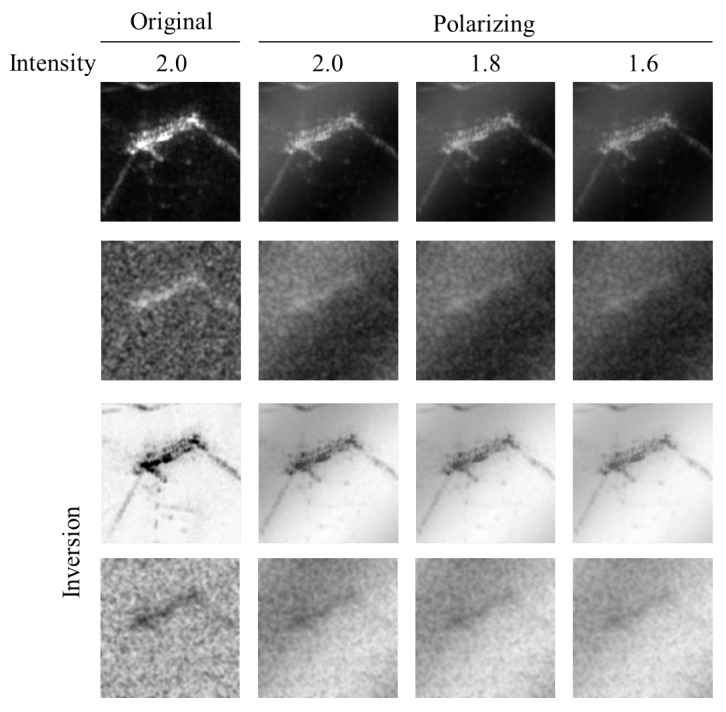
Images depending on whether the polarizing effect is applied or not. We implemented the noising effects on not only by spreading the noise extensively on the original CKI, but also by applying the polarizing effect.

**Figure 5 sensors-20-00094-f005:**
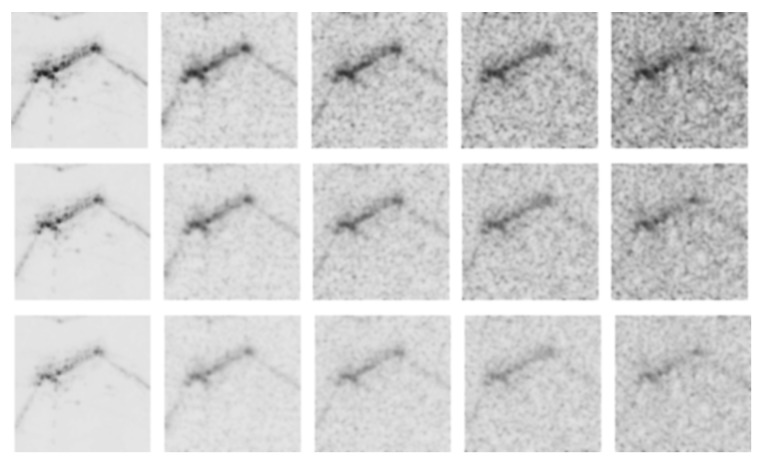
Pixel value inverted images of Figure 3b. The inversion effect occurs differently according to the distance between the sonar sensor and the dummy as like in the TDI-2017 and TDI-2018.

**Figure 6 sensors-20-00094-f006:**
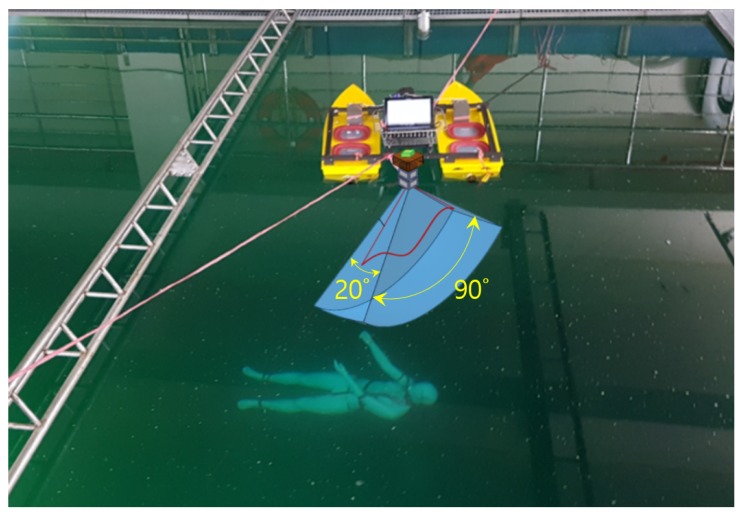
Picture of the experimental setup in the clean water testbed of Korea Institute of Robot and Convergence. BlueView M900-90 has the view field of 90 degrees and the beam width of 20 degrees.

**Figure 7 sensors-20-00094-f007:**
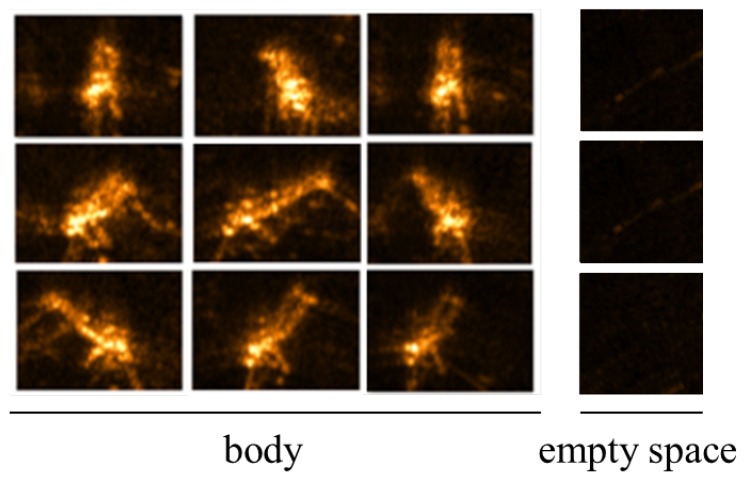
CKI samples including human body and empty space.

**Figure 8 sensors-20-00094-f008:**
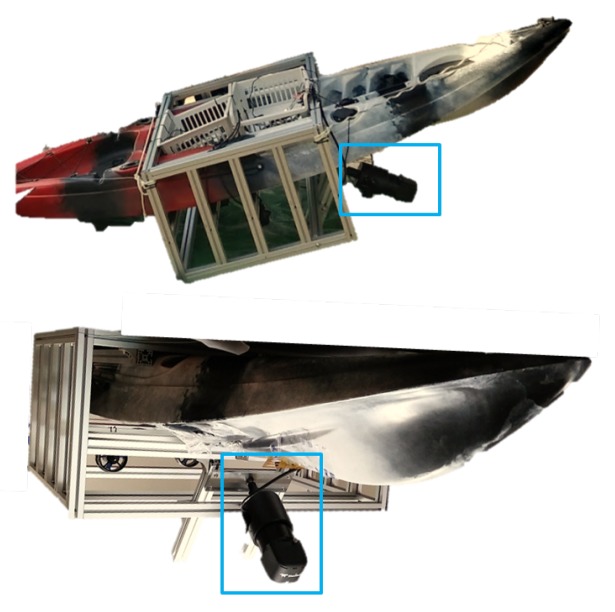
Picture of experimental setup for the turbid water shore of Daecheon beach.

**Figure 9 sensors-20-00094-f009:**
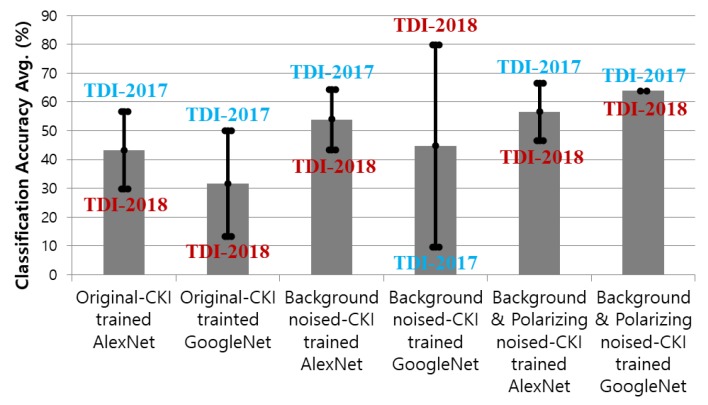
Classification results with TDI-2017 and TDI-2018 by AlexNet and GoogleNet trained with Original-CKI, Background noised-CKI, and Background & Polarizing noised-CKI.

**Figure 10 sensors-20-00094-f010:**
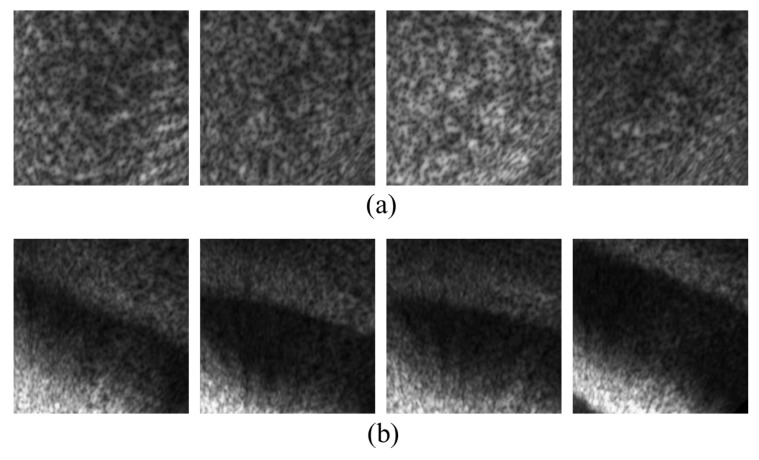
Sample images of (**a**) TDI-2017 and (**b**) TDI-2018 that failed to be classified into the label of background.

**Figure 11 sensors-20-00094-f011:**
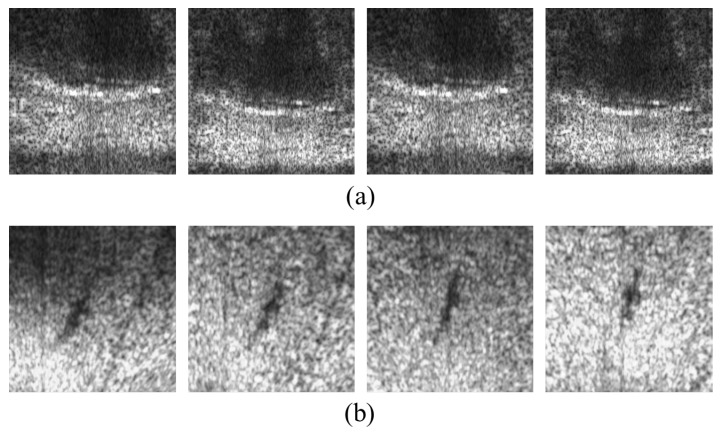
Sample images of (**a**) TDI-2017 and (**b**) TDI-2018 that failed to be classified into the label of body.

**Figure 12 sensors-20-00094-f012:**
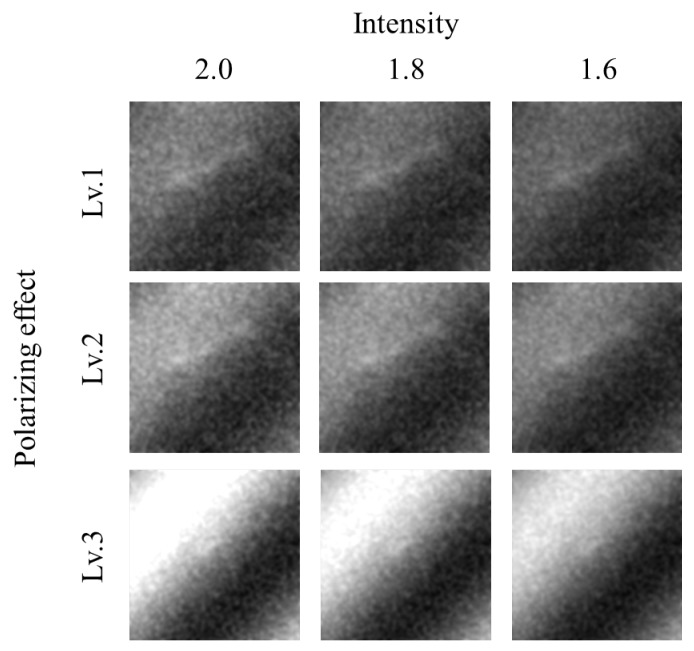
Sample images according to polarizing and intensities.

**Figure 13 sensors-20-00094-f013:**
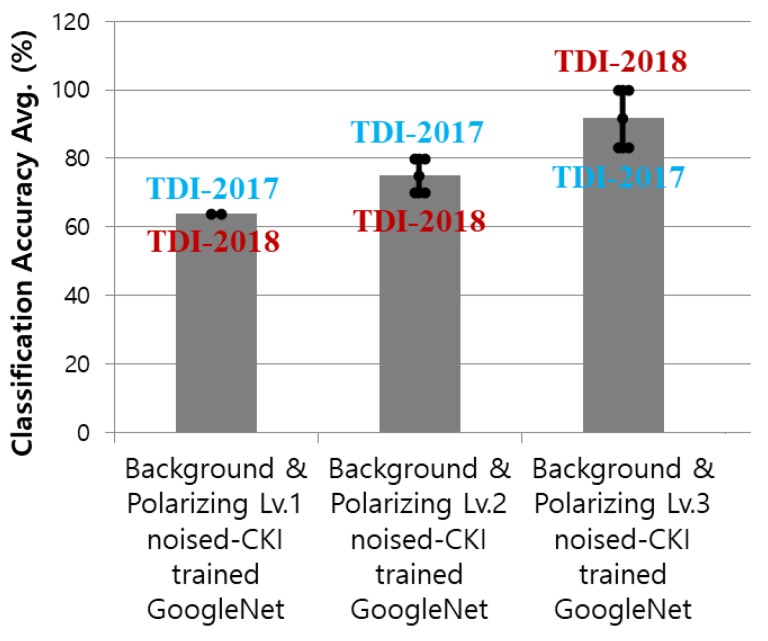
Classification results with TDI-2017 and TDI-2018 by GoogleNet trained with Background & Polarizing noised-CKI according to the polarizing level.

**Table 1 sensors-20-00094-t001:** Classification results through AlexNet and GoogLeNets via Original CKI, Background noised-CKI, and Background & Polarizing noised-CKI.

Models	TDI-2017	TDI-2018	Averages
Original-CKI trained AlexNet	56.6	30	43.3
Original-CKI trained GoogleNet	50	13.3	31.6
Background noised-CKI trained AlexNet	64.4	43.3	53.8
Background noised-CKI trained GoogleNet	9.6	80	44.8
Background & Polarizing noised-	66.6	46.6	56.6
CKI trained AlexNet			
Background & Polarizing noised-	63.8	63.8	63.8
CKI trained GoogleNet			

**Table 2 sensors-20-00094-t002:** Classification results through GoogLeNets according to the polarizing level.

Models	TDI-2017	TDI-2018	Averages
Background & Polarizing Lv.1 noised-	63.8	63.8	63.8
CKI trained GoogleNet			
Background & Polarizing Lv.2 noised-	80	70	75
CKI trained GoogleNet			
Background & Polarizing Lv.3 noised-	83.3	100	91.6
CKI trained GoogleNet

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
