# Peer review of "Study on the Classification Performance of Underwater Sonar Image Classification Based on Convolutional Neural Networks for Detecting a Submerged Human Bodyâ€"

_sensors, 2019, doi:10.3390/s20010094_

Round 1
Reviewer 1 Report
This paper describes a modified Convolutional Neural Network(CNN) for detecting submerged human body based on underwater sonar images. The paper is well written and the results are solid. I suggest minor revisions from authors before publication. Here are my comments.
There are quite a few image based techniques for submerged body detection. However, the reason why choose CNN-based methods needs more explanation.
Regarding the CNN, what is the difference between AlexNet and GoogLeNet?
3. How to determine the failures of classification in figure 10?
Reviewer 2 Report
I strongly suggest to realize in the future a practical application for example in a real situation of first aid action.
Reviewer 3 Report
The article deals with a very important problem of detection of underwater objects. The title of the article is too general because the presented subject matter concerns the introduction of deep learning methods for the detection of objects based on sonar information. The article omits the problem of extraction of objects lying on the bottom. First it is necessary to separate the image of an object from the sonar registration and only then to identify it.
Round 2
Reviewer 3 Report
The revised article is suitable for publication.